# Composition and Physicochemical Properties of Three Chinese Yam (*Dioscorea opposita* Thunb.) Starches: A Comparison Study

**DOI:** 10.3390/molecules24162973

**Published:** 2019-08-16

**Authors:** Haiming Chen, Zhen Hu, Dongli Liu, Congfa Li, Sixin Liu

**Affiliations:** College of Food Science and Engineering, Hainan University, Haikou, Hainan 570228, China

**Keywords:** *Dioscorea opposita* plants, Chinese yam starch, pasting property, freeze–thaw stability

## Abstract

The aim of this work was to compare the composition and physicochemical properties (SEM, XRD, solubility, swelling power, paste clarity, retrogradation, freeze–thaw stability, thermal property, and pasting property) of three Chinese yam (*Dioscorea opposita* Thunb.) starches (CYYS-1, CYYS-2, and CYYS-3) in Yunlong town, Haikou, Hainan Province, China. Our results show that all the CYYS gave a typical C-type X-ray diffraction pattern. The swelling power of CYYS varied from 10.79% to 30.34%, whereas solubility index was in the range of 7.84–4.55%. The freeze–thaw stability of each CYYS showed a contrary tendency with its amylose content. In addition, CYYS-3 showed the highest *T_o_* (81.1 °C), *T_p_* (84.8 °C), *T_c_* (91.2 °C), and *ΔH* (14.1 J/g). The pasting temperature of CYYS-1 increased significantly with sucrose addition. NaCl could inhibit the swelling power of CYYS. There were significant decreases in pasting temperature and pasting time of CYYS when pH decreased.

## 1. Introduction

Starch is an abundant storage polysaccharide in plants and consists of two types of α-d-glucose homopolymers: amylose, a mostly linear or slightly branched molecule of α-d-(1→4)-glucopyranosyl units; and amylopectin, consisting of (1→4)-α-d-glucan short chains linked through α- (1→6) linkages [1,2]. Furthermore, starch granules are stored as part crystalline and part amorphous in structure with various polymorphic types and degrees of crystallinity due to their origins. Starch is widely used in paper, textile, adhesive, sweetener, and food industries and these applications depend on starch properties such as viscosity, swelling volume, solubility, clarity, etc. [3].

Yam, including 600 species, are important root crops cultivated mainly in four distinct centers of origin: the Indochina Peninsula, South China, the Caribbean area, and a location ranging from the West African forest belt to the savannah [4]. *Dioscorea opposite* thumb. is one of the most popular food used in daily life. It has also been used as an important invigorant in traditional Chinese medicine for thousands of years [1]. Despite the high levels of yam production (3.8 × 10^7^ tonnes/year) and high starch content (20–60%, dry basis), Chinese yam starch (CYS) has received little attention from researchers in terms of its industrial potential.

Yunlong, located in Haikou, Hainan, China, a place of the magic red dirt, was created by volcanic eruption tens of thousands of years ago and has been brewing a variety of nutritional microelements. The yams planted in this region have been cultivated mainly in three cultivars (CYYS-1, CYYS-2, and CYYS-3), all of which are rich in selenium and germanium. To date, there have been a few reported studies in these varieties. In addition, salt and sugar are the most important ingredients usually used along with starch in the food industry, the physicochemical properties of starch-based foods often change by the addition of salt and sugar during food processing and storage [5]. Moreover, pH is also an important influencing factor on the properties of starch [6].

The main objective of the present study was to determine and compare the composition and a large set of physicochemical properties (solubility, swelling power, paste clarity, retrogradation, freeze–thaw stability, thermal property, and pasting property) of the starches isolated from the three cultivars (CYYS-1, CYYS-2, and CYYS-3). In addition, the effects of concentration, sucrose, NaCl, and pH on the viscosity parameters of CYYS-1, CYYS-2, and CYYS-3 were also studied. These experiments will help in formulating and proposing new industrial ways for the utilization of this unconventional starch source and adding value to it. In addition to presenting an exhaustive comparison of these cultivars, our study aims to determine the physicochemical properties of CYYS.

## 2. Results and discussion

### 2.1. Compositions

The composition (total starch, amylose, moisture, ash, protein, and fat) of CYYS-1, CYYS-2, and CYYS-3 is summarized in Table 1. The total starch content of CYYS ranged from 69.63 ± 0.83% to 72.78 ± 1.06%, highest for CYYS-2 and lowest for CYYS-3 which was not significantly different from CYYS-1. Amylose contents of CYYS-1, CYYS-2, and CYYS-3 were 20.43 ± 0.25%, 24.81 ± 0.20%, and 21.53 ± 0.11%, respectively. The amylose content of CYYS in this work was significantly higher than that of other cultivars, such as *D. alata* Linn. (18.07%), *D. nipponica* Makino. (17.67%), *D. bulbifera* Linn. (17.61%), and *D. septemloba* Thunb. (13.58%) [7]. The moisture of CYYS in this study was not significantly different (12.26 ± 0.28% to 12.68 ± 0.31%). Ash contents were in the range of 1.75 ± 0.06% to 2.21 ± 0.03%. The isolated starches from CYYS showed low contents of protein and fat, ranging from 0.13 ± 0.01% to 0.20 ± 0.01% and from 0.06 ± 0.01% to 0.08 ± 0.01%, respectively. The different compositions of starches may be related to plant variety, growing zone, and environment [8].

### 2.2. Morphological Properties

SEM micrographs of CYYS-1, CYYS-2, and CYYS-3 with different magnification are shown in Figure 1. Studies reveal that the granules of the three starches present elliptical shapes and have a similar particle size (5–20 μm in diameter). This observation is consistent with previous reports on the shapes of starches from Chinese yam (*Dioscorea opposita* Thunb.) [9,10]. However, these differ from the yam starch (*Dioscorea esculenta*) which shows polygonal shapes with a smooth surface [11]. The granule photos of CYYS-1 and CYYS-2 show smooth surfaces, whereas that of CYYS-3 shows some wrinkles.

### 2.3. XRD

The X-ray diffraction patterns and corresponding crystallinity levels of starches from the three different cultivars are presented in Figure 2. It can be seen that the three starches have a typical C-type crystalline structure, with strong reflections at 2*θ* about 6.8°, 17.7°, 20.2°, and 26.9°. The result is in agreement with those of other *Dioscorea* cultivar starches, which also display the typical C-type X-ray diffraction pattern. The relative crystallinity of the three kinds of starches followed the order: CYYS-3 (34.5%) > CYYS-2 (34.0%) > CYYS-1 (33.9%).

Generally, differences in relative crystallinity between starches can be attributed to the following: (1) crystal size, (2) amount of crystalline regions (influenced by amylopectin content and amylopectin chain length), (3) orientation of the double helices within the crystalline domains, and (4) extent of interaction between double helices [12]. The differences in relative crystallinity of starches can mostly be attributed to amylose content. In previous publications, similar X-ray patterns have been reported for yam starch and its acid-, oxidation-, and enzyme-modified derivatives [13].

### 2.4. Swelling Power (SP), Solubility Index (SI), Paste Clarity, and Retrogradation

The physical properties of the starch paste, such as SP, SI, paste clarity, and retrogradation, greatly influence the application of a particular starch [14]. Most starches are cooked either before or during use. Generally, when starch suspensions are subjected to high temperature, the granules swell quickly and then rupture due to disruption of the amylopectin double helices (dispersed amylose and amylopectin), while amylose preferentially leaches out from the swollen granules [15]. SP and SI of CYYS-1, CYYS-2, and CYYS-3 as a function of heating temperatures ranged from 60 to 100 °C (Figure 3).

SP and SI of all starches were temperature dependent and gradually increased with increasing temperatures from 60 to 100 °C. The SP of different CYYS varied from 10.79% to 30.34%, whereas SI values were in the range of 7.84–14.55% at 100 °C. This result is consistent with the results by Li et al. [13] and Yeh et al. [16]. For CYYS-1, SP and SI increased slightly at temperatures ranging from 60–70 °C, and increased dramatically at temperatures above 70 °C. At temperatures below 80 °C, the SP and SI of CYYS-3 changed minimally, while they increased rapidly over 80 °C. In the case of CYYS-2, SP increased significantly in the whole temperature range (60–100 °C). Moreover, CYYS-3 exhibited the lowest SP and SI values, while the highest SP and SI values were observed for CYYS-2 and CYYS-1, consecutively. According to the results by Muñoz et al. [17] and Sun et al. [18], the SP of corn starch was in the range of 18–20% at 95 °C. Whereas, the SP of cassava starch was ~12% [17]. It has been reported that SP is related to amylose content, the water holding capacity of starch molecules, hydrogen bonding, and crystallinity degree. A low swelling power of CYYS-3 may be attributed to the presence of a large number of crystal regions formed through the association between long amylopectin chains. The solubility suggests that additional interactions may have occurred between amylose–amylose and amylopectin–amylopectin chains [19]. 

When starch suspensions are heated, the granules rupture and the amylose molecules leach out. Upon cooling, the amylose molecules re-associate to form a network during retrogradation. This causes cloudiness in the paste and increase in absorbance [9]. Paste clarity and retrogradation of the investigated starches are shown in Figure 4A,B. As shown in Figure 4A, the clarity of the CYYS gels varied from 1.98% to 15%, and gradually decreased as the storage time increased from 0 to 48 h. The decrease in paste clarity is due to the retrogradation of amylose during storage. The transmittance values of the three starches in this study were significantly lower than those of most yam (*Dioscorea* sp) starches cultivated in the Ivory Coast [4]. The transmittance of the three starches, especially for CYYS-2, was significantly influenced by storage time. Paste clarity exhibited the highest values (15.00–6.68%) for CYYS-2 and the lowest values for CYYS-3 (3.67–1.98%), which can be explained by the results of retrogradation as shown in Figure 4B. Compared to water chestnuts starch, CYYS-1 and CYYS-2 have higher transmittance values. This could be related to the plant variety, growing zone, and environment. In addition, the content and chain lengths of amylose and amylopectin, intra or intermolecular bonding, and lipids have been also reported to be responsible for paste clarity in starches [1]. The retrogradation of the paste was kept nearly constant when storage time was over 12 h (Figure 4B). Furthermore, CYYS-3 showed the highest retrogradation (75.0–82.0 mL/100 mL) when it was constant. This was consistent with the transmittance results (Figure 4A).

### 2.5. Freeze–Thaw Stability

It is well known that when a starch gel is frozen, starch-rich regions are created in the matrix, where water remains partially unfrozen. High solid concentrations in the regions facilitate the association of starch chains to form thick filaments, whereas water molecules coagulate into ice crystals forming a separated phase. These effects contribute to syneresis, which is an important parameter critical to the tendency of retrogradation and the stability of a gel system [20,21]. The freeze–thaw stabilities of starch gels were assessed by determining the level of syneresis of a 6% starch paste following several freeze–thaw and centrifugation cycles. As shown in Table 2, syneresis (%) shows a typical positive correlation with the number of freeze–thaw cycles. This result indicates retrogradation of additional starch formed during the freezing and thawing process. In our test, all the three starches had high syneresis values (40.18–60.23%) after just one cycle and reached maximum after two cycles for CYYS-1 (41.31%) and CYYS-3 (60.52%), and three cycles for CYYS-2 (68.13%), and showed little change through subsequent freeze–thaw cycles. Moreover, the freeze–thaw stability of the three starches followed the order: CYYS-1 > CYYS-3 > CYYS-2. Interestingly, a negative correlation was apparent between the freeze–thaw stability of starches and their amylose contents. As shown in Figure 5, the correlation coefficient between amylose content and syneresis was 0.8589. CYYS-1, which had the lowest amylose content, exhibited the best stability among the three starches. This result was in agreement with a previous study by Jobling et al. (2002) on the potato starch syneresis [14]. In addition, reduction in the average chain length of the amylopectin and various food hydrocolloids was reported, owing to the ability to improve the freeze–thaw stability of starch [6,14,15,20,22,23].

### 2.6. Thermal Property

Melting characteristics of CYYS-1, CYYS-2, and CYYS-3 were examined by differential scanning calorimetry (DSC) under the same conditions. The onset (*T_o_*), peak (*T_p_*), conclusion (*T_c_*), gelatinization temperatures, and enthalpy (*ΔH*) for the endothermic melting of starches varied according to the cultivars (Table 3). There were great variabilities in thermal properties among the three starches. CYYS-3 showed the highest *T_o_* (81.1 °C), *T_p_* (84.8 °C), *T_c_* (91.2 °C), and *ΔH* (14.1 J/g), whereas, CYYS-2 showed the lowest gelatinization temperature (*T_o_* = 67.6 °C and *T_p_* = 73.4 °C), and CYYS-1 required the lowest energy for gelatinization (*ΔH* = 12.3 J/g). Compared to the starches isolated from other plant sources, such as corn, wheat, potato, and cassava starches as reported by Muñoz et al. [21], yam starches in this study showed higher *T_o_*, *T_p_*, and *T_c_* gelatinization temperatures. The differences in gelatinization temperature may be attributed to the differences in amylose content, size, shape, and distribution of starch granules, and to the internal arrangement of starch fractions within the granules [7]. Noda et al. postulated that transition temperatures are influenced by the molecular architecture of the crystalline region, which corresponds to the distribution of the amylopectin short chain (DP, 6~11) [24]. The high gelatinization temperature suggests that the internal network structure of the CYYS granule was dense and may suppress swelling, and CYYS was more thermally stable compared to other normal starches.

The gelatinization enthalpy of yam starches was higher than that of corn starch (11.4 J/g) [18], but lower than that of sweet potato starch (16.5 J/g) [22] and tapioca starch (15.3 J/g) [23]. In general, it is well recognized that the gelatinization enthalpy is related to the number of double helices that unravel and melt during gelatinization [18]. The high *ΔH* of starches suggests that the double helices of the outer branches of adjacent amylopectin chains are strongly associated within the native granule, thus more energy is needed for unravelling and melting during gelatinization. In addition, *ΔH* reflects the quality and quantity of amylopectin crystallites and is an indicator of the loss of molecular order within the granules [25]. That is to say that the molecular architecture of yam starch granules is more easily pyrolyzed compared to those of sweet potato and tapioca starches, but shows better thermal stability than corn starch. Among the three yam starches in this study, CYYS-3 had the best thermal stability. This may be mainly attributed to the different quality and quantity of amylopectin, only retrogradation of which could be quantified by DSC in the assayed temperature range [26].

### 2.7. Pasting Properties

The effect of concentration, sucrose, NaCl, and pH on the viscosity parameters of CYYS-1, CYYS-2, and CYYS-3 are shown in Table 4, which reflects the pasting characteristics of starches during processing and use [27]. When the concentration of starch was 6%, CYYS-2 showed the highest peck viscosity (PV), breakdown (BD), setback (SB), and final viscosity (FV), followed by CYYS-1 and CYYS-3. PV reflects the extent of granule swelling, which is generally caused by weakening of the internal network structure of the granule by gelatinization. This result is in line with the conclusion of swelling power (Figure 3). The BD of the starch paste, defined as the difference between the PV and hot paste viscosity, reflects the stability of the paste during cooking [28]. As shown in Table 4, the low BD value of CYYS-3 (82 BU) indicates that CYYS-3 paste had good stability during cooking. SB shows the viscosity increase upon cooling to 50 °C, indicating the extent of retrogradation of the starch product. FV is a consequence of starch retrogradation, which is related to amylose crystallization, and the different SB and FV values may be attributed to the different amylose contents in the yam starches. In addition, the FV values (2041–2477 BU) at 50 °C were higher than PV values (1572–1957 BU), indicating that the cooked pastes of the three yam starches lack stability.

It is important to study the effect of sugars at a low concentration on starch gelatinization and retrogradation [29]. In this study, three concentrations (3%, 6%, and 9%) of sucrose were used to study their influence on the paste parameters of CYYS-1, CYYS-2, and CYYS-3. Compared to CYYS-1 in the absence of sucrose, the pasting temperature of CYYS-1 yam starch increased significantly with sucrose addition and the degree depended on the sucrose level (Table 4). The observation was in agreement with that for potato starch reported by Chen et al. [28] and corn and wheat starches by Kim and Walker [30]. This was probably caused by sugar hydration, leaving less free water for starch hydration and thus delaying the swelling [30]. The extent of delay was different, depending on the starch source. Adding sucrose enhanced peak viscosity and final viscosity of CYYS-1 and CYYS-3, which were similar to the trends of corn and tapioca starches [31,32]. The result might be attributed to the crosslinking between the sugar molecules and starch chain [29]. In contrast, CYYS-2 showed an opposite trend with a decrease in peak viscosity and final viscosity with added sucrose. Sharma et al. reported the same trends for cassava starch [33]. The breakdown value increased with increasing added levels of sucrose for CYYS-1 and decreased for CYYS-2. The result indicates that sucrose could improve the stability of CYYS-2 paste during cooking, which is mainly related to the hydrogen bonds between starch chain and sucrose. The SB values of yam starches were also affected obviously by sucrose. SB values of CYYS-1 and CYYS-3 increased significantly when sucrose was added. This suggests that sucrose assisted amylose in the process of retrogradation.

The effect of NaCl at four concentrations (2%, 3%, 5%, and 7%) on the pasting properties of CYYS-1, CYYS-2, and CYYS-3 is summarized in Table 4. There were significant increases in pasting temperature and pasting time for all the three starches, indicating that NaCl could inhibit the swelling power of yam starches. The same trend was reported on potato starch by Chen et al. [34]. Strongly hydrated ions can improve the structure of water and inhibit starch gelatinization. Therefore, changes in pasting temperature and pasting time for all three starches may be attributed to the influence of NaCl on the structure of water. A similar trend was also observed with sodium thiosulphate and sodium sulphite [35]. The peak viscosity of the three starch solutions increased significantly by adding NaCl (Figure 3). The changes may be due to a salt–starch interaction, either through the direct binding of salt, or indirectly through the increase in viscosity [5]. Increase in the peak viscosity of maize starch by sodium chloride has been reported and the increase was attributed to the strong starch–salt interaction, suggesting that this interaction leads to reduced Na^+^ mobility for the gelatinized starch leading to higher viscosity. Our results were also similar to those observed for corn and cassava starches [3,5]. Other sodium salts, such as sodium metabisulphite, also increased the peak viscosity when the concentration of salt was below 0.1% [36]. However, sodium hypochlorite, sodium sulphite, and sodium thiosulphate cause a gradual decrease in the viscosity of starch [36,37]. The swollen granules rupture upon continuous heating, resulting in a fall in viscosity (breakdown value). Sodium chloride resulted in elevated breakdown for CYYS-1 and reduction for CYYS-2 and CYYS-3, with increasing concentrations of sodium chloride (2–7%). The result suggests that sodium chloride improved the paste stability of CYYS-2 and CYYS-3, while decreased the paste stability of CYYS-1. The final viscosity of the three starches showed significant increase when the concentration of NaCl increased. The increase in setback values of CYYS-1 and CYYS- 2 upon the addition of NaCl suggests that pastes of CYYS-1 and CYYS-2 exhibit poor stability during cooking. On the contrary, the cooked paste of CYYS-3 showed better stability when NaCl was added.

The effects of pH (4 and 6) on the pasting properties of CYYS-1, CYYS-2, and CYYS-3 are summarized in Table 4. Pasting temperature and pasting time significantly decreased for all the three starches when the pH decreased, indicating that the swelling power of yam starches can be enhanced by acid. In addition, a fall in viscosity (PV and FV) was observed for CYYS-1 and CYYS-2 as the pH decreased. The reduction in peak viscosity at higher acid concentrations may be due to their hydrolyzing action and that formation of radial fissures on starch granules when heated in water. However, the viscosity of CYYS-3 increased as the concentration of acid increased. The difference may be due to the difference in the main chemical composition of the three starches. The breakdown value is a measure of starch fragility [38]. The breakdown values of all the three starches increased significantly when pH decreased from 6 to 4, suggesting that the structure of the starch was badly affected by acid.

## 3. Materials and Methods

### 3.1. Materials

All the chemicals used were of analytical grade. Three varieties of Chinese yam (CYY-1, CYY-2, and CYY-3) were obtained from Yunlong town, located in Haikou, Hainan, China. NaCl and sucrose was purchased from Aladdin Reagent Co., Ltd., Shanghai, China.

### 3.2. Starch Isolation

*D. opposita* Thunb. starch was isolated according to the modified method by Wang et al. [39]. The *D. opposita* Thunb. was peeled and immediately cut into small pieces. The pieces were crushed in a Warring blender (Moulinex, Lyon, France) and suspended in a large excess of distilled water. The slurry was filtered through a 100 μm sieve. When starch was precipitated, the supernatant was removed. The starch was washed three times in distilled water. The slurry containing starch was centrifuged at 3000 *g* for 5 min. The supernatant and upper nonwhite layer, which contained the skin and cell wall, were removed. The white layer (starch layer) was washed three times. Finally, the starch was washed with ethanol. The starch samples were collected and dried overnight at 30 °C.

### 3.3. Morphological Properties

The morphological features of CYYS-1, CYYS-2, and CYYS-3 were observed with an environmental scanning electron microscope (ESEM, Philips XL-3, Holland). Starch samples were suspended in ethanol to obtain a 1% suspension. One drop of the suspension was applied onto an aluminum stub using double-sided adhesive tape and the starch was coated with gold powder to avoid charging under the electron beam after the acetone volatilized. An accelerating potential of 30 kV was used during micrography [40].

### 3.4. XRD

X-ray diffraction patterns of yam starch were analyzed using Rigaku D/max 2500 X-ray powder diffractometer (Rigaku, Tokyo, Japan) with Nickel filtered Cu Kα radiation (k = 1.54056 Å) at a voltage of 40 kV and current of 200 mA. The scattered radiation was detected in the angular range of 3 to 40° (2 h), with a scanning speed of 8°/min and step size of 0.06.

### 3.5. Differential Scanning Calorimetry (DSC)

DSC analyses were performed on a Perkin Elmer DSC 7 device (Perkin Elmer, Norwalk, CT, USA) using sealed stainless-steel pans. The sample pan (10 ± 1 mg of starch and 50 μL of ultrapure water) and the reference pan (50 μL of ultrapure water) were heated from 25 to 120 °C at a scanning rate of 10 °C min^−1^, held for 2 min at 120 °C, and cooled to 60 °C at 10 °C min^−1^. The gelatinization enthalpy (*ΔH*) and the onset (*T_o_*), peak (*T_p_*), and conclusion (*T_c_*) gelatinization temperatures of each sample were then determined on the thermograms. All analyses were performed in duplicate and mean values were calculated.

### 3.6. Swelling Power (SP) and Solubility Index (SI) Determination

SP and SI of CYYS-1, CYYS-2, and CYYS-3 were determined using the method by Mandala and Bayas with some modification [41]. Starch (0.60 g, dry basis) was weighed accurately and transferred quantitatively into centrifuge tubes. Then, deionized water (30 g) was added to the test tube, and the mixture was mixed thoroughly with a Variwhirl mixer for 30 s. The resulting dispersions were heated separately in a water bath at temperatures of 60, 70, 80, 90, and 100 °C for 30 min. The average heating rate, as recorded by thermocouples placed in tubes, was 18 °C min^−1^. During heating, all tubes were covered with plastic covers in order to minimize water loss, and a vortex shaker was applied periodically to prevent granules’ sedimentation. At the end of the heating period, samples were centrifuged (5000 *g*, 15 min) and the precipitate was weighed as W_0_. Both phases were dried at 105 °C for 16 h and the dry solids in precipitated paste (W_1_) and supernatant (W_2_) were calculated. Swelling power is the ratio of the weight of swollen starch granules after centrifugation (g) to their dry mass (g). For each measurement of swelling power and solubility index, 4–8 replicates were used.
Swelling power of starch = W_0_/W_1_(1)
Solubility index of starch = W_2_/W_3_(2)
where W_3_ is the initial weight of starch.

### 3.7. Paste Clarity and Retrogradation Evaluation

The clarity (% transmittance at 650 nm) of starch paste was determined according to the method described by Gul et al. [42]. Starch suspensions (1%, *w*/*w*) was prepared and heated in a boiling water bath for 30 min with constant shaking, and then cooled to room temperature. The light transmittance was read at 650 nm against water blank after the suspension stored for 12, 24, 36, and 48 h.

The retrogradation of the starch paste was determined as follows: 100 mL of starch suspensions (1%, *w*/*w*) was prepared and heated in a boiling water bath for 15 min with constant shaking, and then cooled to room temperature. The paste was then transferred to a graduated measuring cylinder and read the scale of precipitate after stored for 4, 12, 20, 32, 44, 56, and 68 h. The retrogradation of the starch paste was expressed by the precipitate value in starch suspension (mL/100 mL).

### 3.8. Freeze–Thaw Stability

Freeze–thaw stability was determined according to previous methods [43]. Aqueous starch (6%, *w*/*w*) was gelatinized by heating in a boiling water bath with continuous stirring for 20 min. After cooling in a water bath at 25 °C for 120 min, gel samples were divided into three equal parts by weight and then each part was transferred to a centrifuge tube with closed screwcap. The tubes were subjected to cold storage at 4 °C for 16 h to increase nucleation and then frozen at −16 °C for 24 h. To measure freeze–thaw stability, the gels were thawed at 25 °C for 6 h and then refrozen at −16 °C repeatedly up to five cycles. To measure syneresis, samples were centrifuged at 1000 *g* in a refrigerated centrifuge for 20 min after thawing. The supernatant was decanted and weighed. The percentage of syneresis was then calculated as the ratio of the weight of the supernatant decanted to the total weight of the gel before centrifugation and multiplied by 100.

### 3.9. Pasting Properties Determination

The pasting properties of CYYS-1, CYYS-2, and CYYS-3 were measured using a Brabender viscograph-E measurement and control systems (Brabender OHG, Duisburg, Germany). A slurry of starch (6%, *w*/*w*) was stirred manually for 1 min to facilitate dispersion before testing. A 700 cm/g cartridge was fitted with the rotation speed of 75 rpm. The heating and cooling cycles were programmed in the following manner: the samples were heated from 30 to 95 °C (1.5 °C min^−1^), held for 30 min at 95 °C, then cooled to 50 °C at 1.5 °C min^−1^, and finally held for 30 min at 50 °C. The PT, PV, FV, BD, and setback (SB) values are recorded in Table 4.

### 3.10. Statistical Analysis

Analyses of variance were performed, and the mean values ± standard deviations were evaluated by Duncan’s multiplerange test (*p* < 0.05) using SPSS version 13.0 statistical software (SPSS Inc., Chicago, IL, USA). Origin (Origin Lab Co., Pro.8.0, Northampton, MA, USA) software was used for data processing and to create charts.

## 4. Conclusions

The composition and physicochemical properties (SEM, XRD, solubility, swelling power, paste clarity, retrogradation, freeze-thaw stability, thermal property, and pasting property) of CYYS-1, CYYS-2, and CYYS-3 were compared. CYYS-1 and CYYS-2 had a smooth surface, whereas CYYS- 3 showed some wrinkles in the surface. All three starches had a typical C-type crystalline structure. The SP values of CYYS varied from 10.79% to 30.34%, whereas SI values were in the range of 7.84–14.55%. Moreover, the freeze–thaw stability of the three starches followed the order: CYYS-1 > CYYS-3 > CYYS-2. Interestingly, the negative correlation was apparent between the starches’ freeze–thaw stability and their amylose contents. In addition, CYYS-3 showed the highest *T_o_* (81.1 °C), *T_p_* (84.8 °C), *T_c_* (91.2 °C), and *ΔH* (14.1 J/g), whereas, CYYS-2 showed the lowest gelatinization temperature (*T_o_* = 67.6 °C and *T_p_* = 73.4 °C), and CYYS-1 required the lowest energy for gelatinization (*ΔH* = 12.3 J/g). The effect of starch concentration, sucrose, NaCl, and pH on the viscosity parameters of CYYS was also studied. CYYS-2 showed the highest peck viscosity (PV), breakdown (BD), setback (SB), and final viscosity (FV), followed by CYYS-1 and CYYS-3. The pasting temperature of CYYS-1 increased significantly with sucrose addition, and the degree depended on sucrose levels. NaCl could inhibit the swelling power of yam starches. There were significant decreases in pasting temperature and pasting time for all the three starches when pH decreased.

## Figures and Tables

**Figure 1 molecules-24-02973-f001:**
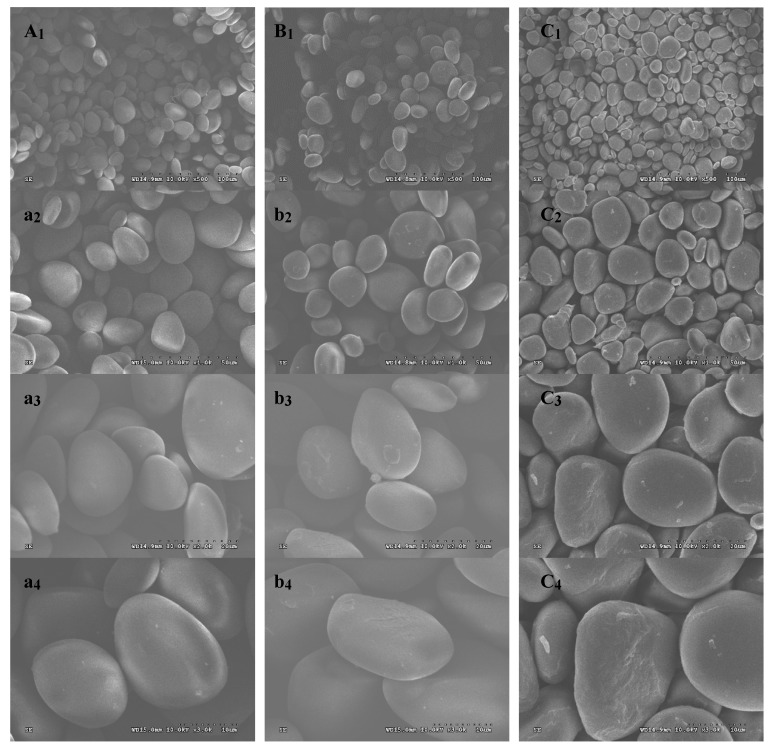
SEM micrographs of CYYS-1 (**A_1_**–**A_4_**), CYYS-2 (**B_1_**–**B_4_**), and CYYS-3 (**C_1_**–**C_4_**). CYYS-1, CYYS-2, and CYYS-3 represent NO. 1-, NO. 2-, and NO. 3-Chinese yam starches in Yunlong town, respectively.

**Figure 2 molecules-24-02973-f002:**
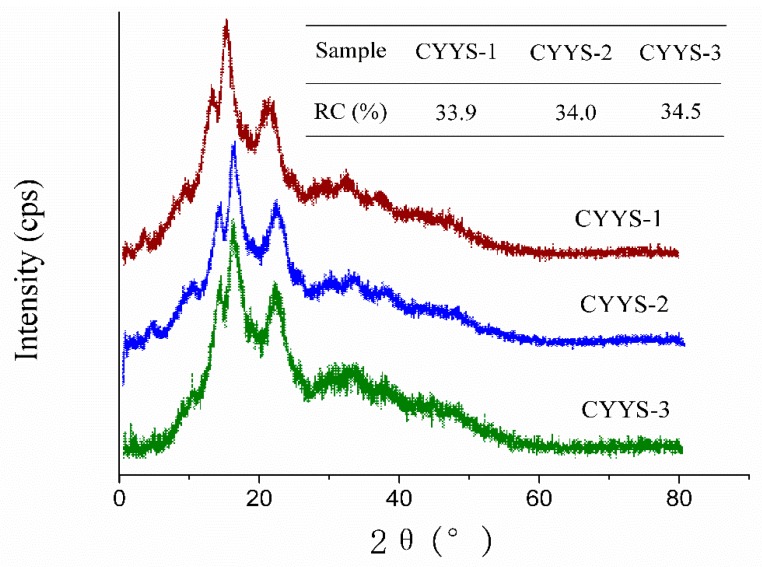
XRD profiles and relative crystallinity (RC) of CYYS-1, CYYS-2, and CYYS-3. CYYS-1, CYYS-2, and CYYS-3 represent NO. 1-, NO. 2-, and NO. 3-Chinese yam starches in Yunlong town, respectively.

**Figure 3 molecules-24-02973-f003:**
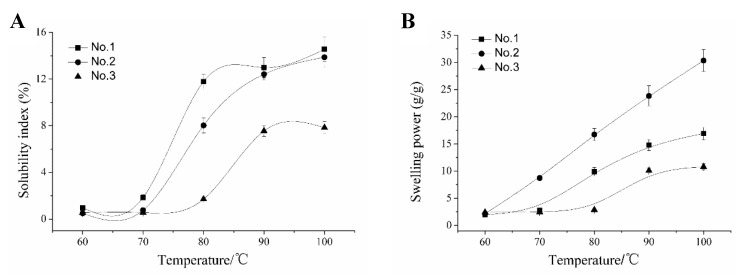
The solubility index (**A**) and swelling power (**B**) of CYYS-1, CYYS-2, and CYYS-3 at different temperatures. CYYS-1, CYYS-2, and CYYS-3 represent NO. 1-, NO. 2-, and NO. 3-Chinese yam starches in Yunlong town, respectively.

**Figure 4 molecules-24-02973-f004:**
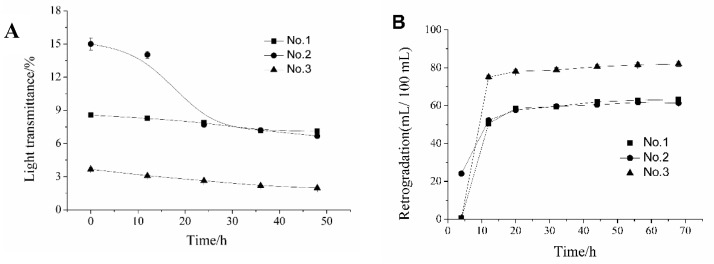
The light transmittance (**A**) and retrogradation (**B**) of CYYS-1, CYYS-2, and CYYS-3 during different storage times. CYYS-1, CYYS-2, and CYYS-3 represent NO. 1-, NO. 2-, and NO. 3-Chinese yam starches in Yunlong town, respectively.

**Figure 5 molecules-24-02973-f005:**
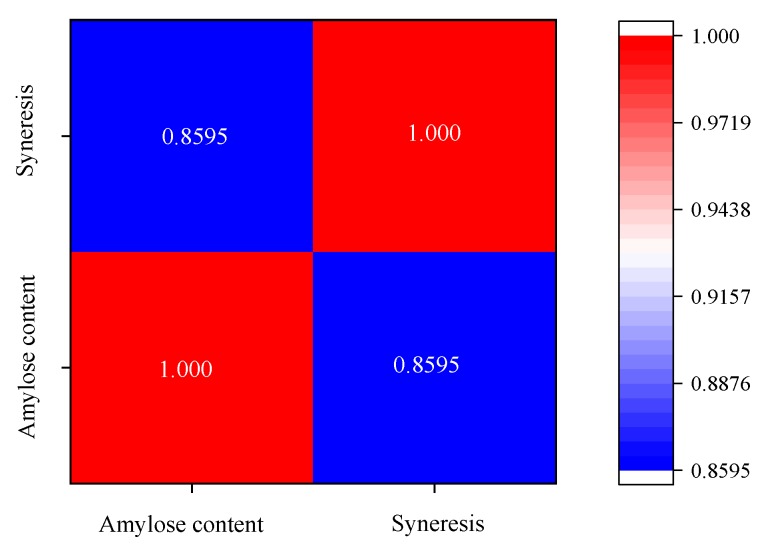
Correlation between Amylose content and syneresis.

**Table 1 molecules-24-02973-t001:** The composition of CYYS-1, CYYS-2, and CYYS-3 ^A^.

Samples	Total starch (%)	Amylose (%)	Moisture (%)	Ash (%)	Protein (%)	Fat (%)
CYYS-1	70.14 ± 0.35 ^b^	20.43 ± 0.25 ^c^	12.26 ± 0.28 ^a^	1.75 ± 0.06 ^c^	0.13 ± 0.01 ^b^	0.06 ± 0.01 ^a^
CYYS-2	72.78 ± 1.06 ^a^	24.81 ± 0.20 ^a^	12.68 ± 0.31 ^a^	2.00 ± 0.02 ^b^	0.19 ± 0.01 ^a^	0.08 ± 0.01 ^a^
CYYS-3	69.63 ± 0.83 ^b^	21.53 ± 0.11 ^b^	12.51 ± 0.24 ^a^	2.21 ± 0.03 ^a^	0.20 ± 0.01 ^a^	0.08 ± 0.01 ^a^

^A^ Assays were performed in triplicate. Mean ± SD values (*n* = 3) in the same row with a different superscript letter are significantly different (*p* < 0.05). CYYS-1, CYYS-2, and CYYS-3 represent NO. 1-, NO. 2-, and NO. 3-Chinese yam starches in Yunlong town, respectively.

**Table 2 molecules-24-02973-t002:** Percentage of water separated (syneresis) of CYYS-1, CYYS-2, and CYYS-3 by freeze–thaw cycling (1–5) ^A^.

Type	Syneresis (%)				
	Cycle 1	Cycle 2	Cycle 3	Cycle 4	Cycle 5
CYYS-1	40.18 ± 0.61 ^a^	41.31 ± 0.45 ^b^	41.38 ± 0.38 ^b^	41.44 ± 0.39 ^b^	41.53 ± 0.21 ^b^
CYYS-2	60.23 ± 1.04 ^a^	64.74 ± 1.21 ^b^	68.13 ± 0.68 ^c^	68.98 ± 0.83 ^c^	69.06 ± 0.88 ^c^
CYYS-3	58.36 ± 0.43 ^a^	60.52 ± 0.81 ^b^	61.37 ± 1.13 ^b^	61.39 ± 1.00 ^b^	61.55 ± 0.75 ^b^

^A^ Assays were performed in triplicate. Mean ± SD values (*n* = 6) in the same row with a different superscript letter are significantly different (*p* < 0.05). CYYS-1, CYYS-2, and CYYS-3 represent NO. 1-, NO. 2-, and NO. 3-Chinese yam starches in Yunlong town, respectively.

**Table 3 molecules-24-02973-t003:** Thermal properties of CYYS-1, CYYS-2, and CYYS-3.

Samples	*T_o_* (°C)	*TP* (°C)	*TC* (°C)	ΔH (J/g)
CYYS-1	68.4 ± 0.1 ^b^	73.8 ± 0.2 ^b^	82.4 ± 0.1 ^c^	12.3 ± 0.2 ^c^
CYYS-2	67.6 ± 0.4 ^c^	73.4 ± 0.1 ^c^	85.1 ± 0.2 ^b^	12.9 ± 0.3 ^b^
CYYS-3	81.1 ± 0.2 ^a^	84.8 ± 0.2 ^a^	91.2 ± 0.3 ^a^	14.1 ± 0.5 ^a^

*T_o_*, *T_p_*, *T_c_*, and *ΔH* are onset, peak, conclusion temperatures, and enthalpy, respectively. Assays were performed in triplicate. Mean ± SD values in the same column with a different superscript letter are significantly different (*p* < 0.05). CYYS-1, CYYS-2, and CYYS-3 represent NO. 1-, NO. 2-, and NO. 3-Chinese yam starches in Yunlong town, respectively.

**Table 4 molecules-24-02973-t004:** The RVA parameters of CYYS-1, CYYS-2 and CYYS-3 ^A^.

Samples	PV (BU)	TV (BU)	BDV (BU)	FV (BU)	SBV (BU)	P-Time (min)	P-Temp (°C)
CYYS-1 (6%)	1729	1354	375	2179	825	4.67	80.2
CYYS-2 (6%)	1957	1581	376	2477	896	5.13	81.8
CYYS-3 (6%)	1572	1490	82	2041	551	6.53	85.4
CYYS-1 + 3% sucrose	1873	1444	429	2272	828	4.73	81.00
CYYS-1 + 6% sucrose	1995	1510	485	2354	844	4.73	81.05
CYYS-1 + 9% sucrose	2154	1633	521	2519	886	4.80	81.75
CYYS-2 + 3% sucrose	1830	1554	276	2432	878	5.20	82.60
CYYS-2 + 6% sucrose	1878	1585	293	2505	920	5.20	82.55
CYYS-2 + 9% sucrose	2063	1704	359	2675	971	5.20	83.45
CYYS-3 + 3% sucrose	1692	1616	76	2204	588	6.33	86.00
CYYS-3 + 6% sucrose	1871	1778	93	2410	632	6.40	86.70
CYYS-3 + 9% sucrose	1941	1864	97	2420	656	7.00	88.40
CYYS-1 + 2% NaCl	1938	1767	121	2964	1197	5.40	85.85
CYYS-1 + 3% NaCl	2063	1931	132	3198	1267	5.60	86.75
CYYS-1 + 5% NaCl	2219	2066	153	3453	1387	5.93	88.40
CYYS-1 + 7% NaCl	2344	2259	185	3675	1416	6.20	88.35
CYYS-2 + 2% NaCl	1705	1578	127	2646	1068	6.00	86.75
CYYS-2 + 3% NaCl	1711	1684	27	2707	1023	6.13	88.40
CYYS-2 + 5% NaCl	1829	1769	60	2890	1121	6.47	89.05
CYYS-2 + 7% NaCl	1932	1909	23	3046	1137	6.53	89.90
CYYS-3 + 2% NaCl	851	738	120	1026	311	6.77	91.60
CYYS-3 + 3% NaCl	912	792	113	1103	288	6.89	93.30
CYYS-3 + 5% NaCl	939	839	100	1108	269	7.13	94.00
CYYS-3 + 7% NaCl	1098	1003	95	1250	267	7.28	94.90
CYYS-1 (pH = 4)	1667	1262	405	2011	749	4.67	80.15
CYYS-1 (pH = 6)	1783	1572	211	2651	1079	5.07	81.80
CYYS-2 (pH = 4)	1614	1035	579	1542	507	5.00	80.85
CYYS-2 (pH = 6)	1674	1431	243	2247	816	5.13	81.65
CYYS-3 (pH = 4)	1601	1514	87	2053	539	6.53	85.85
CYYS-3 (pH = 6)	1043	957	86	1290	333	7.00	86.70

^A^ CYYS-1, CYYS-2, and CYYS-3 represent NO. 1-, NO. 2-, and NO. 3-Chinese yam starches in Yunlong town, respectively.

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
