# Peer review of "Composition and Physicochemical Properties of Three Chinese Yam (Dioscorea opposita Thunb.) Starches: A Comparison Study"

_molecules, 2019, doi:10.3390/molecules24162973_

Round 1

Reviewer 1 Report

the reserach presented is interesting to scientific community, however, some improvements are suggested as follows:

literature citig in the text should be ut before the dot, eg. [1]. instead . [1] in all tables and figures, please add full names of analysed starches, not only used abbrev. in tab. 1 - if the values are presented for starch and not for tuber, ash content is very high - please explain this. in some places (eg. line 93), lit. reference is in superscript some properties are compared to corn, potato and tapioca starches, some are compared to other types. you should focus on starches with granules of similar shape and size as yours. correlation between amylose content and freeze-thaw stability was not determined statistically - you could add this (not only for this analysis, but for others as well). it could be interesting to determine the correlation with cristallinity as well. the gelatinisation temperature is rather high compared to other normal starches - explain why. based on what did you discuss thermal stability in section of DSC analysis? line 224 - is crosslinking really a possible interaction between sucrose and starch? please explain how and refer to literature data based on what did you choose such high salt concentration for analysis?  when determining SP and SOL you state you dried samples for 16 h. why this time? why not drying until constant mass was reached? could this influence results obtained? when determining pasting properties, you used 6% starch suspension. you should always make suspensions based on dry matter (6% of dry matter, not total starch mass), since difference in starch concentrations will influence results. brabender viscograph gives results in brabender units - how did you recalculate them to cP? conclusions should be improved. where is applicability of analysed starches? what are advantages and what are disadvantages compared to commercial starches?

Author Response

The reserach presented is interesting to scientific community, however, some improvements are suggested as follows:

1) Literature citing in the text should be ut before the dot, eg. [1]. instead . [1]

Response: It has been revised as your suggestion.

2) In all tables and figures, please add full names of analyzed starches, not only used abbrev.

Response: The full names of the analyzed starches has been added in all tables and figures according to your suggestion.

3) In Tab. 1 - if the values are presented for starch and not for tuber, ash content is very high - please explain this.

Response: The data have been rechecked and the data are accurate. The high content of ash may be related to the Dioscorea species. According to the previous report (Food Hydrocolloids 29 (2012) 35-41), the ash contents of the starch isolated from Dioscorea L. species were 1.71% (D. bulbifera Linn.) and 1.92% (D. septemloba Thunb.). In addition, the ash content of yam (Dioscorea opposite Thunb) starch was up to 2.18% as reported by Xia et al. (2011) (Starch/Sta¨ rke 2011, 63, 616–624).

Reference

Jiang, Q., Gao, W., Li, X., Xia, Y., Wang, H., & Wu, S., et al. (2012). Characterizations of starches isolated from five different Dioscorea L. species. Food Hydrocolloids, 29(1), 35-41.

Xia, L., Wenyuan, G., Qianqian, J., Yanli, W., Xinhua, G., & Luqi, H.. (2011). Physicochemical, crystalline, and thermal properties of native, oxidized, acid, and enzyme hydrolyzed Chinese yam (Dioscorea opposite Thunb) starch. Starch - Stärke, 63(10), 616-624.

4) In some places (eg. line 93), lit. reference is in superscript

Response: It has been revised according to your suggestion (line 95).

5) Some properties are compared to corn, potato and tapioca starches, some are compared to other types. you should focus on starches with granules of similar shape and size as yours.

Response: Some properties of CYYSs were compared with starches with granules of different shape and size. This study compared the characterizations of starches among CYYS-1, CYYS-2 and CYYS-3, and also with other species. The aim of this study was to discuss whether it is possible to replace other starches with CYYS and to expand its application scope.

6) Correlation between amylose content and freeze-thaw stability was not determined statistically - you could add this (not only for this analysis, but for others as well). it could be interesting to determine the correlation with crystallinity as well.

Response: The correlation between amylose content and freeze-thaw stability was determined and showed in Fig. 4. While, the correlation coefficient between amylose content and crystallinity was low (0.1250).

7) The gelatinization temperature is rather high compared to other normal starches - explain why.

Response: The differences in gelatinization temperature may be attributed to the differences in amylose content, size, shape and distribution of starch granules, and to the internal arrangement of starch fractions within the granules. The high gelatinization temperature suggested that the internal network structure of the CYYS granule was dense and may suppress swelling, and CYYS was more thermal stable when compared to other normal starches.

8) Based on what did you discuss thermal stability in section of DSC analysis?

Response: The thermal stability was discussed on the base of the onset (To), peak (Tp), conclusion (Tc) gelatinization temperatures, and enthalpy (ΔH) for endothermic melting of starches.

9) line 224 - is crosslinking really a possible interaction between sucrose and starch?

Response: According to the results in our manuscript and the data reported previous (Kohyama, & Nishinari, 1991), the crosslinking between the sugar molecules and the starch chain may be a possible cause.

10) Please explain how and refer to literature data based on what did you choose such high salt concentration for analysis?

Response: The range of the salt concentration was chosen on the base of the literature data.

Reference:

Bircan, C., and S. A. Barringer. (1998) "Salt‐Starch Interactions as Evidenced by Viscosity and Dielectric Property Measurements." Journal of Food Science.6:983-986.

Jyothi, A. N., Sasikiran, K., Sajeev, M. S., Revamma, R., & Moorthy, S. N.. (2005). Gelatinisation properties of cassava starch in the presence of salts, acids and oxidising agents. Starch - Stärke, 57(11), 547-555.

Chen, H. H., Wang, Y. S., Leng, Y., Zhao, Y., & Zhao, X. (2014). Effect of NaCl and sugar on physicochemical properties of flaxseed polysaccharide-potato starch complexes. Science Asia, 40(1), 60-68.

Oosten, B. J. (1982). Tentative Hypothesis to Explain How Electrolytes Affect the Gelatinization Temperature of Starches in Water. Starch - Stärke, 34(7), 233–239. doi:10.1002/star.19820340706

11) When determining SP and SOL you state you dried samples for 16 h. why this time? why not drying until constant mass was reached? could this influence results obtained?

Response: To obtain a constant mass, 14~16 h was needed for all the samples. Therefore, we used 16 h as the standard value.

12) When determining pasting properties, you used 6% starch suspension. you should always make suspensions based on dry matter (6% of dry matter, not total starch mass), since difference in starch concentrations will influence results.

Response: Weight in our manuscript refers to dry weight. And it has been revised accordingly (line 215, 332, 343).

13) Brabender viscograph gives results in brabender units - how did you recalculate them to cP?

Response: All the units in our manuscript have been rechecked and corrected (Table 4) (Line 221, 225).

14) Conclusions should be improved. where is applicability of analysed starches? what are advantages and what are disadvantages compared to commercial starches?

Response: Conclusion has been improved and the possible application of CYYS has been added according to your suggestion (line 369-386). Thanks for your careful review.

Reviewer 2 Report

This study investigated the composition and physicochemical properties of starches from three Chinese yam. Overall, this manuscript contains some novel points, and more importantly, it can provide critical scientific evidence for their food applications in the future. The writing of the manuscript is generally excellent, with clear logic and presentation, but please finally check it for possible grammar or typing errors.

Author Response

Response: Thanks for your careful review and the overall manuscript has been rechecked carefully by AJE company.